# A Case-Finding Protocol for High Cardiovascular Risk in a Primary Care Dental School—Model with Integrated Care

**DOI:** 10.3390/ijerph20064959

**Published:** 2023-03-11

**Authors:** Amazon Doble, Raul Bescos, Robert Witton, Shabir Shivji, Richard Ayres, Zoë Brookes

**Affiliations:** 1Faculty of Health, University of Plymouth, Plymouth PL4 8AA, UK; 2School of Health Professions, University of Plymouth, Plymouth PL4 8AA, UK; 3Office of the Chief Dental Officer, London SE1 6LH, UK

**Keywords:** hypertension, hyperglycaemia, hypercholesterolaemia, case-finding, health referral

## Abstract

Background: National Health Service (NHS) strategies in the United Kingdom (UK) have highlighted the need to maximise case-finding opportunities by improving coverage in non-traditional settings with the aim of reducing delayed diagnosis of non-communicable diseases. Primary care dental settings may also help to identify patients. Methods: Case-finding appointments took place in a primary care dental school. Measurements of blood pressure, body mass index (BMI), cholesterol, glucose and QRisk were taken along with a social/medical history. Participants with high cardiometabolic risk were referred to their primary care medical general practitioner (GP) and/or to local community health self-referral services, and followed up afterwards to record diagnosis outcome. Results: A total of 182 patients agreed to participate in the study over a 14-month period. Of these, 123 (67.5%) attended their appointment and two participants were excluded for age. High blood pressure (hypertension) was detected in 33 participants, 22 of whom had not been previous diagnosed, and 11 of whom had uncontrolled hypertension. Of the hypertensive individuals with no previous history, four were confirmed by their GP. Regarding cholesterol, 16 participants were referred to their GP for hypercholesterolaemia: 15 for untreated hypercholesterolaemia and one for uncontrolled hypercholesterolaemia. Conclusions: Case-finding for hypertension and identifying cardiovascular risk factors has high acceptability in a primary dental care setting and supported by confirmational diagnoses by the GP.

## 1. Introduction

National Health Service (NHS) health screening in the United Kingdom (UK), was introduced in 2009 to reduce cardiovascular disease (CVD) risks and adverse event occurrence [1]. Subsequently, national reviews have highlighted the need to maximise prevention by improving cardiovascular risk case-finding coverage and outputs, while making every patient contact count [2]. Since that time, other organisations such as pharmacies and community health programs have implemented preventative cardiovascular case-finding in non-clinical settings, in order to reduce the morbidity and mortality of CVD and ease pressure on the NHS system [3]. Cardiovascular case-finding has had varying degrees of success within the UK [2,4,5], and this may relate to the variability of measurements taken in non-clinical settings [6]. Early diagnosis and preventative treatment of high blood pressure (hypertension), including identifying undiagnosed patients, could avoid 9710 heart attacks and 14,500 strokes, potentially saving the NHS £274 million per year [7]. Similarly, undiagnosed high cholesterol levels (hypercholesterolemia), are recognised as a contributing factor to coronary artery disease and stroke, with nearly 8 million adults in the UK currently taking lipid-lowering drugs, such as statins, to reduce the risk of developing these complications after being diagnosed [8]. Additionally, undiagnosed and uncontrolled high blood glucose (hyperglycaemia) can lead to many severe complications, including but not limited to kidney, heart, peripheral vascular system and eye damage [9]. Thus, it is vital to manage hyperglycaemia effectively to prevent disease states and improve patient outcomes [10]. However, according to research conducted by the British Heart Foundation, 11.4% of the current UK population is at high risk of a major cardiovascular event (e.g., stroke, heart attack) due to non-controlled hypertension, hypercholesterolemia and/or hyperglycaemia [11].

Hypertension is known as the ‘silent killer’ [12], and this emphasises the importance of case-finding patients as a preventive strategy to reduce the risk of major cardiovascular events (e.g., stroke). Dental practices are good settings in which to identify people with untreated hypertension because a substantial number of people, 36.9% of the UK adult population, have attended an NHS dentist within the last 24 months [13], either for cleaning or treatment. The dental environment provides a unique site for obtaining accurate cardiometabolic health measurements, given the abundance of clinically trained staff and emergency medical supplies, whilst operating under strict patient confidentiality and data protection. They are also equipped for referral to other primary care providers, such as general practitioners (GPs), and other secondary care services, including cancer care. Being in a medical environment instils confidence in patients when receiving results and advice [14]. This is an advantage over some other non-medical environments which have been used previously, such as shopping malls [6].

There is also positive evidence demonstrating higher patient acceptance of participating in cardiovascular case-finding at the dentist [14,15]. A study by Creanor et al. (2014) assessed patients’ attitudes towards diabetes and cardiovascular screening within a dental school setting, concluding that 83% of patients attending student dental clinics would be willing to participate in health assessments at the dentist [14]. Similarly, Sproat et al. (2009) measured the blood pressure of 114 individuals visiting the dentist in the UK, and found that 39% of them had either a systolic reading greater than 140 mmHg, or a diastolic reading greater than 90 mmHg, or both [15]. This highlights the potential of undertaking health assessments in non-traditional settings in the UK.

To the best of our knowledge, a dental school setting has not been previously used to case-find patients with cardiovascular risk factors including high blood pressure, cholesterol, and glucose, with the inclusion of GP referral and robust follow up investigations [16].

This study has thus developed an applied model of cardiometabolic health measurements within the University of Plymouth Dental School, in collaboration with Peninsula Dental Social Enterprise (PDSE) primary care clinics, serving an area of low socioeconomic status [17]. Collaboration with PDSE provides a good opportunity to recruit participants that may not be sufficiently health conscious to self-refer themselves for health screening at the GP [18,19,20]. Indeed, the area of Devonport that PDSE dental school serves is currently ranked the second most deprived ward in Plymouth (2nd out of 20), and is within the <1% most deprived neighbourhoods in the UK [17]. Thus, this protocol will also be facilitating patient access to care for some of the most at risk groups in the UK for management of cardiovascular risk factors including hypertension, a current NHS priority [21].

The main aims of this study were to use a dental school primary care setting, due to the large volume of patients routinely seen, to (i) establish case-finding clinics to identify people with high cardiometabolic risk factors; and (ii) establish a dental-GP referral network for systemic patient health.

## 2. Materials and Methods

### 2.1. Ethical Aspects

This study was approved by the Human Ethics Committee of the University of Plymouth (reference number: 2684). The protocol and design comply with all ethical standards outlined by the responsible committee on human research experimentation.

### 2.2. Data Recording

A unique code was given for each patient to maintain confidentiality within the data storage. Restricted and confidential clinical data were recorded and stored long term in the dental software system, on password-protected computers behind restricted swipe card access in PDSE.

### 2.3. Participants and Recruitment

Participants were recruited into the study through the dental school onboarding routes (Figure 1), including:Triage sessions, in which people without a current dentist are assessed by dental school staff for their suitability as patients for dental school students. These sessions were set up and supported by clinical staff members of the Peninsula Dental School/PDSE.Student/Staff clinics, which invited participants already undergoing treatment at the dental school to book an appointment for our case-finding clinic.Phone recruitment, which recruited participants who were missed at triage clinics or on the waiting list for triage clinics to optimise participant uptake. This involved sending participants the information sheet via email or letter after discussing the study with them via phone.

These pathways are outlined in Figure 1.

Participants were booked into a case-finding clinic a minimum of one week later, and the patient’s routine dental care at the dental school continued separate to this appointment. Trial staff provided each participant with an information sheet and answered any questions participants had. All participants within the study were recruited from PDSE clinic facilities in the Plymouth area, including Devonport Dental Education Facility.

Patients recruited to the trial were over 40 years of age, which is in agreement with NHS health screening initiatives [1] and given the increased risk of hypertension development in this age group [22]. All patients, irrespective of age and acceptance at dental triage, received routine dental care separate from the case-finding pilot. However, after recruitment, two patients were excluded as they were found to be <40 years of age. No other exclusion criteria were applied for participation in case-finding. However, full social and medical histories were taken to assess previous history of cardiovascular risk factors and social determinants.

### 2.4. Applied Model

The applied model for cardiometabolic health measurements within a dental school environment is outlined in Figure 2; however, a further explanation of this can be seen below.

Participants were taken to a separate clinic room within the dental school. Measurements were all taken by either dentists, dental nurses or research scientists trained on the same protocols for standardisation and overseen by a clinical lead.

A medical, dental, and social history was also taken using a health questionnaire, including any previous diagnosis of hypertension, hypercholesterolaemia, stroke, heart attack, or diabetes, and a social history including their level of education and current occupation. Participants were then asked to lay down on a medical bed in a supine position for 10 min, before systolic (SBP) and diastolic blood pressure (DBP) were recorded in the left arm. Three measurements were taken, with an average of the lowest two values being taken.

After this, body height and weight were measured using a stadiometer and mechanical scale, respectively, to calculate the body mass index (BMI). Finally, blood cholesterol and glucose were measured using a portable electronic device (Accutrend Plus, [23]) to assess cardiovascular risk using the Qrisk [24,25]. The Qrisk was used to support any referral for blood pressure, glucose, or cholesterol to their GP. We reviewed patient feedback after the first 100 patients, and broadened the case-finding initiative with the addition of blood glucose measurements. A glucose reading was only taken from participants who had arrived 3 h fasted, as per our information sheet.

The case-finding clinic was conducted separately from dental appointments to minimise the impact of any dental anxiety when taking blood pressure readings [26].

### 2.5. Referral and Follow Up

On completion of the case-finding clinic, those participants under high cardiovascular risk, including those with a previous medical history or current medication use for the risk factor (high levels of hypertension, hyperglycaemia or hypercholesterolaemia (outlined in Section 2.6) were provided with a letter including their assessment data, and they were strongly encouraged to visit their GP for diagnostic purposes and management. Patients were also offered community self-referral services, such as Plymouth’s Livewell self-referral services [27], for management of diet, weight loss, anxiety, smoking cessation and alcohol use. Additionally, a letter was sent to the participants’ GPs from the clinic if they were found to have hypertension, hyperglycaemia or hypercholesterolaemia, in accordance with NICE guidelines [7] (Figure 3). Furthermore, a cholesterol-lowering plan leaflet from Heart UK, a cholesterol charity, was provided to participants presenting with hypercholesterolemia [28].

Patients were then followed up at 2.5-week intervals until such time as they had either visited their GP for confirmation of the potential hypertension/hypercholesterolaemia diagnosis, received any lifestyle or medication intervention outside the GP setting, or decided to take no action on our health findings.

### 2.6. Identification and Management of Variables

Participants were classified into categories of cardiovascular health as outlined below. According to blood pressure levels, participants were classified hypertension if they presented at the hypertension case-finding clinic with an SBP over 140 mmHg and/or DBP over 90 mmHg, following current NICE hypertension guidelines [29], irrespective of whether they were using anti-hypertensive medication. Participants with an SBP of over 180 mmHg and/or DBP over 120 mmHg were asked to contact their GP on the same day, and if unable to do this, to attend an A&E unit, following NICE hypertension guidelines [29].

Using cholesterol measurements, participants were classified as having healthy cholesterol (<5 mmol/L), high cholesterol (5–6 mmol/L) or extremely high cholesterol (>6 mmol/L), as per NICE guidelines [30], irrespective of whether they are using statin medication. Cholesterol, blood pressure and health questionnaire answers were used to calculate a Qrisk3 for each participant, which provided the patient with information on their risk of developing a heart attack or stroke within the next 10 years [24]. Participants were classified as low risk (Qrisk3 < 10%) or high risk (Qrisk > 10%) as per NICE guidelines, using the website to input and calculate these scores contemporaneously [25].

With glucose measurements, participants within the range of 4–7 mmol/L were classified as healthy, but values were only considered if patients arrived at the appointment after 3 h of fasting, in order to conform to NHS health check procedures [31,32]. Glucose measurements above 7 mmol/L were classified as hyperglycaemic, as per NICE guidelines [33,34]. However, these measurements were supportive rather than diagnostic at this stage, due to lack of consistency with fasting.

For BMI measurements, participants were classified as overweight (BMI between 25 and 29.9) or obese (BMI between 30 and 39.9) [35], with BMI calculated at the time using height and weight measurements input into the NHS BMI calculator [36].

## 3. Results

Participant recruitment and attendance is shown in Figure 4, and the population’s characteristics are displayed in Table 1. Two participants were excluded early in the trial, as the age range for the study was decreased for the reasons outlined in Section 2.3. These two participants, represented in Figure 4, did receive care and a case-finding appointment; however, their data have been excluded from our report.

Of the 182 people that booked a hypertension case-finding appointment, 123 (67.5%) attended their appointment; however, only 121 were included in our assessment due to age limitations. Hypertension was detected in thirty-three (27.3%) attendees, and consequently, they were referred to their GP. Twelve participants with high blood pressure not diagnosed previously followed up with their GP; however, only ten undertook further monitoring under the GP’s direction. Four were diagnosed with hypertension. Furthermore, eleven (9.1%) of the participants that were detected with high blood pressure were already using anti-hypertensive medication.

Blood cholesterol was measured in fifty-three (43.8%) participants. Sixteen of them (30.0%) were referred to their GP for presenting with hypercholesterolemia, including one (1.9%) participant on statin medication for high cholesterol. Five participants with suspected untreated hypercholesterolemia followed up with their GP; however, only three undertook further monitoring under the GP’s direction. Of those who followed up with their GP and undertook further monitoring, one was successfully diagnosed with hypercholesterolemia without a previous diagnosis.

The Qrisk of the twenty-three participants (43.4%) that underwent cholesterol measurements was over 10%. Furthermore, the BMI of forty-three participants (35.5%) of this study was >25 (overweight), and forty-two participants (34.7%) had a BMI >30 (obesity). Twenty-five (20.7%) participants were current smokers and twelve (9.9%) were former smokers.

Of the 123 participants, only ten (8.1%) underwent glucose case-finding due to equipment delays and technical difficulties. Of these ten, one (10.0%) presented with suspected untreated hyperglycaemia. We are still awaiting communication from this participant regarding whether they followed up with their GP after referral. No participants were identified as having uncontrolled hyperglycaemia.

## 4. Discussion

The current study successfully describes a functional and applied protocol within a UK dental school setting for measuring blood pressure, BMI, cholesterol, and glucose, combined with medical history taking. In turn, it has been able to successfully establish a hypertension case-finding clinic for CVD within a primary care dental setting, thus allowing the identification of people with increased cardiometabolic risk factors. The participants in this study would not otherwise have attended for health assessments, as they attended the dental clinics with medical histories perceived as free of cardiovascular disease or with CVD under control with medication.

The proportion of patients with untreated hypertension was higher than the national average of 12% [37]. This may be a result of operating within an area of lower socioeconomic status, as rates of hypertension are higher in this population [38]. Additionally, we successfully identified 11 (9.1%) individuals as having uncontrolled hypertension, which is higher than the 2019 NHS national average of 5% [37]. However, after communication regarding GP follow-up and further monitoring, four individuals with suspected untreated hypertension were confirmed with a diagnosis by their GP, representing 3.3% of our overall population. We believe that the confirmed diagnosis would be higher if more than 54.5% of those referred for untreated hypertension had followed up with their GP. We are the first case-finding study in a dental setting that we are aware of that has followed up with the GP in this way. Considering the extra care taken with accuracy in our clinical setting, the reduced incidence of referral demonstrates the importance of integrated care in any case-finding model, and may explain the lack of success previously reported with some hypertension health screening approaches [6,15,39]. Ideally, we would also undertake ambulatory blood pressure monitoring to further improve the accuracy of the measurement with which we refer to the GP, and this will form the basis of future studies.

Additionally, a large proportion of our population were obese (34.7%); this figure again was higher than the national average of 28% [40]. This supports our assumption of the lack of health-conscious individuals in our clinic location. However, our protocol identified that 13.2% of our participants had possible untreated and uncontrolled hypercholesterolemia, which is less than the national average of 43% for hypercholesterolemia [37]. However, only 43.8% of our population received a cholesterol reading, due to factors such as technical difficulties or blood/needle phobia on the day. Perhaps this result would be higher and in line with the national average if we had been able to measure blood cholesterol in the whole population of this study.

Overall, 38% of our population were referred to their GP due to cardiometabolic variables that may increase the risk of cardiovascular disease; this figure is higher than that quoted by the British Heart Foundation, who state that that 11.4% of the current UK population is at high risk of a major cardiovascular event [11]. Given that 20.7% of our population were current smokers, which is slightly higher than the national average of 16% [37], and that a larger amount of our population were obese, there appears to be a greater need for further lifestyle counselling and education in a range of settings, which should sit alongside smoking cessation programs.

To our knowledge, this study is the first to establish a cardiovascular risk case-finding environment within a UK dental school. Overall, 82.9% of people approached for the study took up the offer of an appointment for cardiovascular risk case-finding; our findings remain consistent with the study by Creanor et al. (2014) which reported that almost 83% of people attending student dental clinics were willing to participate in health assessments at the dentist [14]. Here, moving from perceived acceptance to actual attendance, 67.5% of participants then attended their booked appointment, supporting known difficulties with attending the dentist in areas of lower socioeconomic status. Additionally, our study identified 27.3% of participants with increased blood pressure, whereas Sproat et al. (2009)’s study within a dental practice was able to identify 39%. However, their study did utilise participants aged 18 and over, which may be something to consider for further studies [15]. The study by Sproat et al. (2009) also advised participants to contact their GP [15], whereas here, we actively referred the participant to their GP and actively confirmed diagnoses, with data to evidence this.

These case-finding clinics have established a dental–GP referral network for systemic patient health within an area of social deprivation, carrying out health assessments within a population at higher risk of CVD. It was considered that working interprofessionally with GPs to ensure accurate case-findings was essential to the success of this program. The cardiovascular health assessments performed in the dental setting appeared to be accurate, as participants had their potential diagnoses confirmed by the GP, whereas other studies did not report on such referrals [15,41]. If case-finding is not performed as carefully and accurately as possible at the dentist, there is a risk of burdening of GPs with false-positive results, which this study has attempted to avoid by describing this referral process.

This protocol also fostered the dental school’s collaboration with the Office of the Chief Dental Officer in the UK and the CORE20PLUS5 programme to influence national and local policy and improve links between oral and systemic health, including care pathways. As many dental schools are positioned close to areas of socio-economic deprivation [42], in the future, it may be that dental schools will be ideal settings for providing access to oral health services to people in deprived areas in which CVD is more common, reaching some of the most at-risk populations. These data have been fed into an existing hypertension case-finding program within the dental school to work towards the next step of commissioning, which is beyond the scope of this study.

However, this study has some limitations which need to be considered. For example, it only included participants over 40 years old, in order to assess those most at risk of CVD; however, younger individuals may be less vulnerable, but nevertheless are also susceptible to cardiometabolic complications if left without assessment [22]. Future research trials would benefit from the inclusion of participants over the age of 18, not just over 40, to encourage individuals to manage their health from a younger age. Furthermore, participants were placed in the supine position when taking the blood pressure; our method, can therefore be viewed as having lower blood pressure readings compared to studies done on the sitting position, which could be something else to consider in future studies [43]. Additionally, this study continues to be run in conjunction with other research studies which have provided sufficient clinical and non-clinical trial staff to be available to run the clinics. If dental schools or primary care dental practices wanted to run these clinics outside of a trial environment, they would be required to find funding, equipment and human resources. This study began to test for fasting glucose levels; indeed, others have performed similar studies, measuring blood glucose at the dentist [44]. It was interesting that here, we found a patient with uncontrolled diabetes despite the small population, and that patients wanted this additional measurement to be performed. Our future studies will continue glucose testing and adapt the method further, with, for example, HbA1c monitoring and 3-month average glucose levels, to try and avoid false-positive and false-negative glucose results, similar to our aim with blood pressure [31,45]. Additionally, despite all precautions being undertaken within our protocol to remove white-coat syndrome, when taking blood pressure readings, there is the possibility that dental anxiety may still be present at opportunistic blood pressure assessments within a dental healthcare setting [26]. Furthermore, 24 h or 1-week ambulatory monitors could be given as part of the trial to reduce false positives and aid in confirming diagnoses. Lastly, given the pressure on NHS GPs within the UK currently [46], with our network referring at-risk participants to their GP, we must determine whether GP surgeries have the capacity needed to deliver appointments to these referred individuals.

## 5. Conclusions

In conclusion, this study is the first to successfully establish a case-finding protocol for people with high cardiovascular risk within a UK dental school. This intervention can help to reduce the risk of major cardiovascular events and their cost to the NHS.

## Figures and Tables

**Figure 1 ijerph-20-04959-f001:**
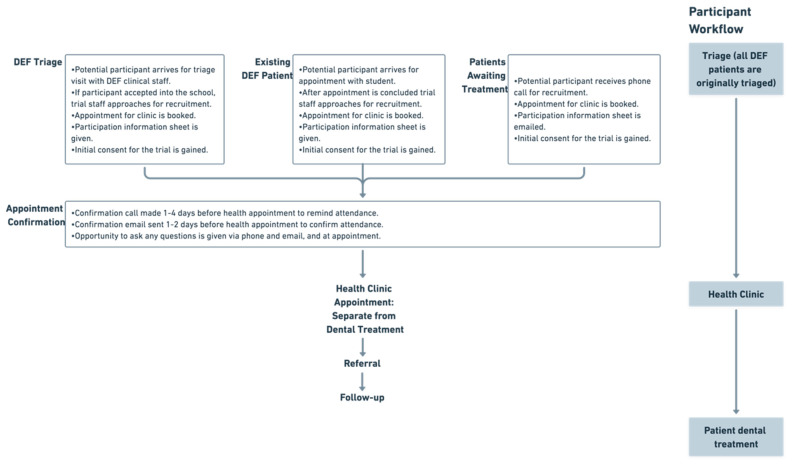
Pathway for recruitment of participants for hypertension case-finding within Peninsula Dental School. DEF—dental education facility.

**Figure 2 ijerph-20-04959-f002:**
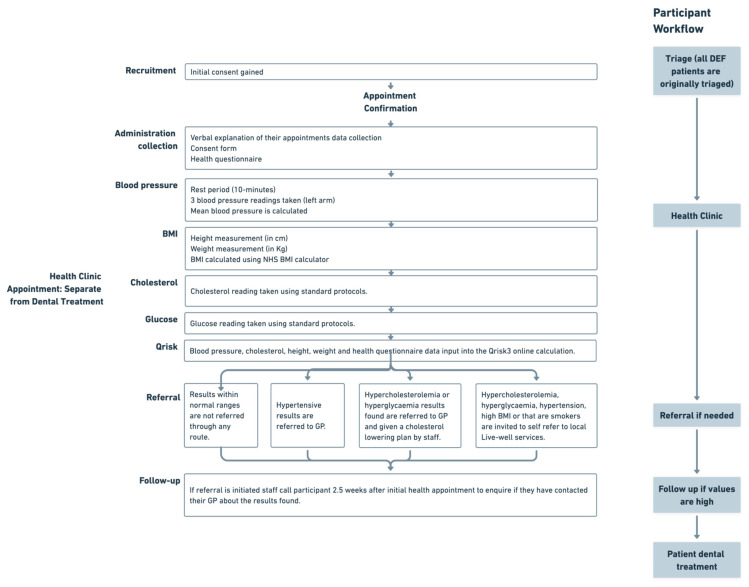
Applied workflow method for cardiovascular risk case-finding within a dental school, including a referral and follow-up pathway. DEF—dental education facility, GP—general practitioner, BMI—body mass index.

**Figure 3 ijerph-20-04959-f003:**
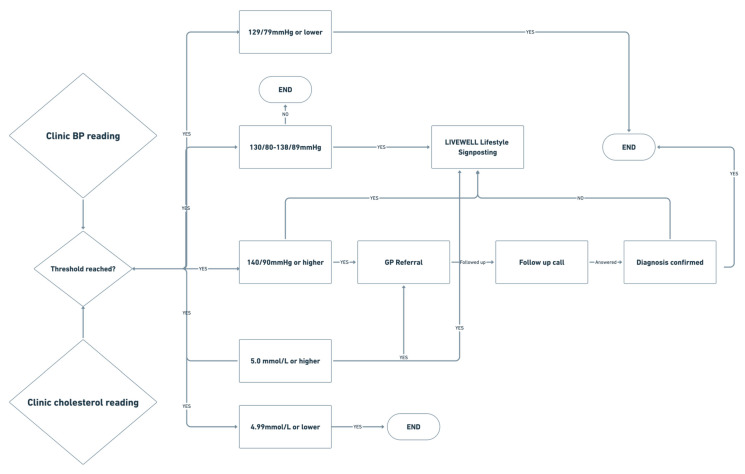
Referral pathways involving the GP and support services after participants received their blood pressure and cholesterol measurements. GP—general practitioner, BP—blood pressure.

**Figure 4 ijerph-20-04959-f004:**
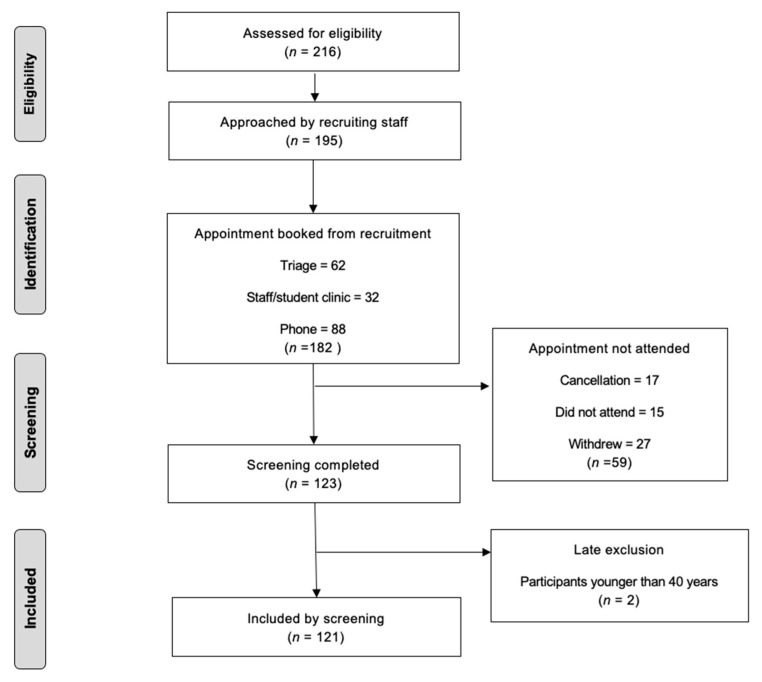
Consort figure displaying the total number of participants that completed a return case-finding appointment at Peninsula Dental School (*n* = 123) following their recruitment at an earlier appointment at least 1 week previously (*n* = 182).

**Table 1 ijerph-20-04959-t001:** Population characteristics, (mean ± SD) to 1 d.p. BMI—body mass index, BP—blood pressure, SD—standard deviation.

Characteristic	All Participants (*n* = 121)	Hypertensive Untreated (*n* = 22)	Hypertensive Uncontrolled (*n* = 11)	Hypercholesterolemia Untreated (*n* = 15)	Hypercholesterolemia Uncontrolled (*n* = 1)
*Mean (±SD)*					
Age (years)	57.3 ± 10.4	58.0 ± 11.4	60.6 ± 10.4	53.3 ± 10.7	59 ± 0
BMI (kg/cm)	28.7 ± 6.2	28.2 ± 5.3	33.6 ± 5.9	29.9 ± 5.4	32.4 ± 0
Systolic BP (mmHg)	130.6 ± 17.1	148.6 ± 16.4	155.5 ± 12.3	129.6 ± 12.9	137 ± 0
Diastolic BP (mmHg)	79.0 ± 9.2	89.4 ± 8.3	84.6 ± 8.5	79.1 ± 8.1	78 ± 0
*n (%)*					
Gender					
Female	71 (58.7%)	9 (40.9%)	6 (54.5%)	10 (66.7%)	0 (0.0%)
Male	50 (41.3%)	13 (59.1%)	5 (55.5%)	5 (33.3%)	1 (100%)
Smoking Status					
Never	84 (69.4%)	7 (31.8%)	8 (72.7%)	10 (66.7%)	1 (100%)
Former	12 (9.9%)	1 (4.5%)	2 (18.2%)	2 (13.3%)	0 (0.0%)
Current	25 (20.7%)	4 (18.2%)	1 (9.1%)	3 (20.0%)	0 (0.0%)
Vape Use (Never smokers)	2 (1.7%)	1 (4.5%)	0 (0.0%)	0 (0.0%)	0 (0.0%)
Vape Use (Former smokers)	4 (3.3%)	0 (0.0%)	1 (100%)	1 (100%)	0 (0.0%)
Race					
White	118 (97.5%)	22 (100%)	11 (100%)	15 (100%)	1 (100%)
Other (Chinese, Mixed Race)	3 (2.5%)	0 (0.0%)	0 (0.0%)	0 (0.0%)	0 (0.0%)
BMI					
Underweight	4 (3.3%)	1 (4.5%)	0 (0.0%)	0 (0.0%)	0 (0.0%)
Healthy	32 (26.4%)	4 18.2%)	0 (0.0%)	4 (26.7%)	0 (0.0%)
Overweight	43 (35.5%)	8 (36.4%)	5 (45.5%)	5 (33.3%)	0 (0.0%)
Obese	42 (34.7%)	9 (40.9%)	6 (54.5)	6 (40.0%)	1 (100%)
Education					
School	54 (44.6%)	10 (45.5%)	5 (45.5%)	7 (46.7%)	1(100%)
Collage	17 (14.0%)	2 (9.1%)	0 (0.0%)	2 (13.3%)	0 (0.0%)
University or Higher	34 (28.1%)	8 (36.4%)	3 (27.3%)	6 (40.0%)	0 (0.0%)
No answer	16 (13.2%)	2 (9.1%)	3 (27.3%)	0 (0.0%)	0 (0.0%)
Diabetes (all type II)	15 (12.4%)	1 (4.5%)	4 (36.4%)	1 (6.7%)	0 (0.0%)
Heart attack	3 (2.5%)	0 (0.0%)	0 (0.0%)	1 (6.7%)	0 (0.0%)

## Data Availability

Contact A.D. or Z.B. for data related to this study.

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
