# Peer review of "A Case-Finding Protocol for High Cardiovascular Risk in a Primary Care Dental School—Model with Integrated Care"

_ijerph, 2023, doi:10.3390/ijerph20064959_

Round 1

Reviewer 1 Report

The authors managed to demonstrate the potential of a simple idea, which may be useful in cardiovascular prevention field. The manuscript is well written, and I don't have significant issues to mention.

I would ask if data related to uncontrolled or potential newly diagnosed diabetes is available to be added to results section. As the title refers to general CV risk, I think adding data about glycemia in the text and not only the focus on hypertension or cholesterol, will fit even more the main message of the trial. 

Author Response

Thank you for your comments on our manuscript. We hope that the following will address your useful feedback.

We designed this study originally as a blood pressure case finding initiative, but found that our patients were interested in broader health screening, and not all arrived fasted. So yes we have begun to do this, but at present we have only done case finding for hyperglycaemia in 10 participants, finding 1 patient (10%) as untreated for hyperglycaemia. We also only were able to take measurements when people arrived fasted (Line 154). However, we have updated the manuscript cautiously to present this information, as we agree it may be of interest to the reader and fit nicely into the main message of the trial.

- On line 153-155 we have included the method:

We reviewed patient feedback after the first 100 patients, and broadened the case finding initiative with the addition of blood glucose measurements.

- On line 167-169 we have included the method:

Furthermore, a cholesterol lowering plan leaflet from Heart UK The Cholesterol Charity, was provided to participants presenting with hypercholesterolemia.

- On Lines 239-243 we have included the results:

Of the 123 participants, ten (8.1%) underwent glucose case finding during the latter part of the study. Of the 10 patients assessed, one (10%) presented with suspected untreated hyperglycemia. We are still awaiting communication from this participant regarding whether they followed up with their GP after referral. No participants were identified as having uncontrolled hyperglycemia.

- On Lines 333-339 we have included in the Discussion

This study began to test for fasting glucose levels; indeed others have performed similar studies measuring blood glucose at the dentists [44]. It was interesting here, to find a patient with uncontrolled diabetes, despite the small population, and that patients wanted this additional measurement. Our future studies will continue glucose testing and adapt the method further, with for example HbA1c monitoring and 3-month average glucose levels, to try and avoid false positive and negative glucose results, similar to our aim with blood pressure [31, 45].

Reviewer 2 Report

This study examined the use of case finding appointments to identify and refer patients at risk for cardiovascular disease. The novelty of this study the use of established dental-GP referral networks, which may not be feasible in other systems with more fragmented care. Of the 182 patients that agreed to participate over a 14-month period, 123 attended, 33 presented with high blood pressure and 16 were referred for hypercholesterolemia.

Major comments:

1.       In the introduction, the authors state on lines 73 and 74 “to the best of our knowledge a dental school setting has not been previous used to find patients with cardiovascular risk factors. It appears that the authors are aware of at least some of the other work that uses dental settings for screening as they reference one study immediately prior to this sentence. Perhaps the authors were trying to state this is the first to provide an actual referral, rather than only informing the patient? The following article is one example that utilized a dental school.

 https://doi.org/10.14219/jada.archive.2007.0268

2.       The authors stated that they measured glucose. However, I’m unclear as to why the authors developed the protocol that was used? They state that a glucose reading was only taken from participants who had arrived 3 hours fasted. What was the rationale for this approach? Current guidelines have a random glucose cut-off, that is to say no need for the patient to be fasting. There is also the option of testing an A1c so that fasting is not a consideration. Despite the methodological limitations with this approach, I don’t see any report of the screening glucose level results anywhere in the manuscript?

3.       I’m unclear why age was not considered prior to scheduling this appointment? What happened to the patients that arrived and were too young? Did they still receive care?

4.       You state that those with elevated cholesterol levels are given a cholesterol lowering plan by staff. I did not find a more detailed description of this in the text. Can you please provide a more detailed explanation. Is someone in the clinic developing an individualized plan? Is this just standard heart healthy diet information? Who is providing this education?

Minor comments:

Reference 29, I’m unclear why you are citing a guidelines for type 1 (shouldn’t this be type 2)

Author Response

Major comments:

  1. In the introduction, the authors state on lines 73 and 74 “to the best of our knowledge a dental school setting has not been previous used to find patients with cardiovascular risk factors. It appears that the authors are aware of at least some of the other work that uses dental settings for screening as they reference one study immediately prior to this sentence. Perhaps the authors were trying to state this is the first to provide an actual referral, rather than only informing the patient? The following article is one example that utilized a dental school.

Thank you for this point, we have now updated the relevant lines 74 to 75 to reflect that we are talking about ‘actual referral’ as being the more unique part of our study:

To the best of our knowledge, only one to dental school setting has been used previously to case find patients with cardiovascular risk factors including high blood pressure, cholesterol, and glucose [16], and we may be the first to further include GP referral and more robust follow up investigation .

The suggested Greenberg reference as [16] in the text so thank you for the https://doi.org/10.14219/jada.archive.2007.0268

  1. The authors stated that they measured glucose. However, I’m unclear as to why the authors developed the protocol that was used? They state that a glucose reading was only taken from participants who had arrived 3 hours fasted. What was the rationale for this approach? Current guidelines have a random glucose cut-off, that is to say no need for the patient to be fasting. There is also the option of testing an A1c so that fasting is not a consideration. Despite the methodological limitations with this approach, I don’t see any report of the screening glucose level results anywhere in the manuscript?

We ask the participants to arrive 3 hours fasted, as we have developed the protocol in this way to stay in line with the NHS health check, which when testing for diabetes most commonly uses fasted glucose levels [31. 32]. With the aim of the project being to make our measurements as accurate as possible and avoid producing false positives for the GPs, this was one of the rationales for this approach.

A1c measurement, 3 month glucose measurement and referral to the GP for possible hyperglycaemia are all options for our future studies to further improve our accuracy and is therefore now included in lines 341-347 of the discussion:

This study began to test for fasting glucose levels; indeed others have performed similar studies measuring blood glucose at the dentists [44]. It was interesting here, to find a patient with uncontrolled diabetes, despite the small population, and that patients wanted this additional measurement. Our future studies will continue glucose testing and adapt the method further, with for example HbA1c monitoring and 3-month average glucose levels, to try and avoid false positive and negative glucose results, similar to our aim with blood pressure [31, 45]

We have also added some more information to the methods and more information to the results to explain in more detail how blood glucose measuring came about and what was found (reviewer 1 asked a similar question):

- On line 153-155 we have included the method:

We reviewed patient feedback after the first 100 patients, and broadened the case finding initiative with the addition of blood glucose measurements.

- On line 167-169 we have included the method:

Furthermore, a cholesterol lowering plan leaflet from Heart UK, a charity which supports lifestyle interventions to reduce cholesterol.

- On Lines 239-243 we have included the results:

Of the 123 participants, ten (8.1%) underwent glucose case finding during the latter part of the study. Of the 10 patients assessed, one (10%) presented with suspected untreated hyperglycemia. We are still awaiting communication from this participant regarding whether they followed up with their GP after referral. No participants were identified as having uncontrolled hyperglycemia.

  1. I’m unclear why age was not considered prior to scheduling this appointment? What happened to the patients that arrived and were too young? Did they still receive care?

We approached over 40s in line with other NHS health screening initiatives [1]. Over 40s were therefore not recruited into this case finding initiative. They did however receive routine dental care, as this ran separately to the case finding clinics. Therefor the text in lines 215-219 was change to:

Patients recruited to the trial were over 40 years of age, in agreement with NHS health screening initiatives [1] and given the increased risk of hypertension development in this age group [22]. All patients irrespective of age and accepted at dental triage received routine dental care separate to the case finding pilot. However, after recruitment two patients were excluded as they were found to be <40 years of age.

  1. You state that those with elevated cholesterol levels are given a cholesterol lowering plan by staff. I did not find a more detailed description of this in the text. Can you please provide a more detailed explanation. Is someone in the clinic developing an individualized plan? Is this just standard heart healthy diet information? Who is providing this education?

We apologise for this lack of information. The cholesterol lowering plan provided is an educational leaflet from Heart UK The cholesterol charity. This has now been outlined in the manuscript and the plan referenced in lines 177-179 of the manuscript:

Furthermore, a cholesterol lowering plan leaflet from Heart UK The Cholesterol Charity, was provided to participants presenting with hypercholesterolemia [28].

Minor comments:

Reference 29, I’m unclear why you are citing a guidelines for type 1 (shouldn’t this be type 2)

Thank you for pointing out this error. The text has now been amended, using the  current NICE guidelines, which refer to type 2 diabetes with reference [34]:

  1. NICE. Type 2 diabetes in adults: management: NICE; 2022 [Available from: https://www.nice.org.uk/guidance/ng28.

Round 2

Reviewer 2 Report

Thank you for adequately addressing all of my comments and recommendations.